# Developments in Chinese Attitudes to Animal Welfare

**DOI:** 10.3390/ani15060878

**Published:** 2025-03-19

**Authors:** Clive J. C. Phillips

**Affiliations:** 1Sustainability Policy (CUSP) Institute, Curtin University, Kent St., Bentley, Perth 6102, Australia; clive.phillips58@outlook.com or clive.phillips@curtin.edu.au; 2Institute of Veterinary Medicine and Animal Science, Estonia University of Life Sciences, Kreutzwaldi 1, 51014 Tartu, Estonia

**Keywords:** China, animal welfare, gender, legislation, religion

## Abstract

China is the world’s largest producer of farm animals; therefore, attention to Chinese attitudes to animal welfare has major significance from a global perspective. Their cultural attitudes towards animals have traditionally been strongly influenced by religion, but, more recently, this has been changing in response to the emergence of an empathetic approach to animals, as evidenced by growing attention by scientists, governmental officials, and activists. In addition, the law, gender, age, experience and occupation influence attitudes. The public’s desire to have food of the highest quality is sometimes equated with good animal welfare. In contrast to the approach in the West of removing farm animals from cages and giving them more space, scientific approaches in China tend to focus on remote sensing of animal welfare and technological solutions to welfare problems, which is more feasible in this period of rapid intensification of animal production.

## 1. Introduction

China has seen momentous social changes in the last one hundred years, which have influenced the attitudes of its people towards animals, but its long history has also had a major influence. Animal production systems in this country are of worldwide significance as it is in the top three countries for the volume of production of all the major meat categories [1]. It also has more meat consumers than any other country—79% of its 1.4 billion inhabitants, or over 1.1 billion people [2]. It is also the largest producer of aquatic products in the world, producing 35% of the global output [3]. However, the prospect of cellular meat and synthetic milk is growing and may be strongly supported in China now that the technologies have been proven in laboratories [4]. Acceptability of these products is uncertain: in one study, the acceptance of cultured products (and plant-based alternatives), including willingness to pay more, was relatively low [5], whereas in another, the acceptance of cultured meat was high [6]. If these products succeed in capturing the Chinese market, they could change Chinese attitudes toward animals radically.

The aim of this narrative review is to explore the historical versus recent developments in animal welfare in China and examine whether Chinese attitudes towards animal welfare have changed. Initially, the method of the study and the historical context of animals’ role in China and people’s attitudes toward them are presented, followed by a focus on the attitudes of Chinese people working in the livestock industries. Then, the factors influencing attitudes to animals in China are reviewed, including the law, type of production system, gender, age, experience, occupation, political ideology, and religion. Finally, the changing state of attitudes toward animals in China is considered, and conclusions are drawn about attitudes toward animals in the future, including the scenario of cultured animal products becoming popular. Throughout, there is a focus on farm animals, with some consideration of wildlife.

## 2. Method

The Web of Science, Google Scholar, and personal contacts in PR China (henceforth China) were used to initially extract relevant articles, using the keywords ‘animal’ and (‘China’ or ‘Chinese’) and (‘welfare’ or ‘wellbeing’) and (‘attitude’ or ‘public’). The Web of Science search produced 2270 hits, which was refined to 43 hits with the compulsory inclusion of ‘attitude’. All of these articles were inspected for relevance, and full papers were read if appropriate.

## 3. Animals in Chinese Society in the Last Millennium

For most of the last thousand years, large livestock, including horses, camels, buffalo, cattle, and donkeys, were widely used in agricultural production systems. Many were used for transport purposes. Horses were extensively used for military purposes, and their widespread value in society led to them holding a high moral status in the view of everyday Chinese society. Breeding and nutrition of livestock became established as scientific disciplines, with controlled reproduction and animal care being managed statewide [7]. Hunting was an important part of daily life for China’s imperial rulers in the 18th century [8], to the extent that deer populations were diminished as a result. Common people were more likely to engage in falconry.

In agriculture, livestock have played a much more important role in the extreme environments (mountains, marshes, steppes, and deserts) than in the lowlands, where the pressure of large populations ensured that crop production prevailed. In the medieval period, large herds and flocks were established by invaders, mostly in the north and west, but later, small-scale farming systems became the main form of land use, preferred by the government as they empowered the agrarian society with their own land [9]. Pigs and smaller livestock came to form an integral part of the society, to whom the modern Chinese owe their heritage. In more recent times, many smallholdings have been eliminated by the government’s desire to concentrate farm animals into enormous enterprises, even by world standards [10]. Dairy farms are some of the largest in the world, with up to 30,000 cows. In part, this was prompted by a desire to attain self-sufficiency in clean and safe food production from animals after some disastrous incidents, such as the milk contamination scandal [11]. This scandal was created because milk was deliberately contaminated with melamine, a potentially toxic compound rich in nitrogen that was added to milk to artificially increase the apparent protein content. Tens of thousands of people were hospitalized, mostly children under the age of three, and six babies died.

Increasingly rapid growth in the production of food from animals has been evident since the late 1970s, with a mean 8% annual growth rate in meat production [11]. Memories of food shortages were still alive. Nowadays, some of the world’s largest and most intensive farm animal production systems are in China in a drive to maintain high levels of food safety and efficient use of scarce resources, such as animal feed, land, and water.

## 4. Historical Context of Animal Welfare in Chinese Society

Animal welfare has played a strong part in Chinese literature, mythology, and poetry, as it has in the West [12]. Traditional culture recognizes compassion, empathy, and respect for all living beings [12], described in more detail under Religion. However, recent Chinese cultural development has been vastly different from that of Western countries, such as the USA, UK, and Australia [13,14]. First, Chinese people have a strong sense of hierarchy and respect for authority, accepting inequality between those with and without power in society. Second, they have a much stronger allegiance to the community, whereas those in Western societies tend to work independently. Lastly, Chinese people tend to have more of a sense of long-term orientation and engage less in impulsive reactions, generally showing restraint in their dealings with others [14]. Although the Chinese public has little understanding of the modern Western concept of animal welfare [12], there is recognition of the Five Freedoms by some zoo staff, at least [15]. At a recent conference on animal welfare in China, there was strong support for the Five Freedoms and ‘the important value of animal welfare in sustainable agri-food systems’ [16].

Animal science has been growing in China, with a focus on pigs and poultry [6]. Remote sensing of animal behavior and welfare is strongly featured [17]. With increased attention to animal welfare science, Chinese scientists have been attempting to encourage Chinese farmers to give greater consideration to animal welfare [18]. However, few address the fact that causing animals pain is fundamentally wrong, which Cao [19] believes has resulted in an insensitive attitude toward animals being prevalent in Chinese society. Cao argues that the respect for authority imbued by Confucianism has led to a hierarchical societal structure in which animals are not respected as much as they are in the West. They are used extensively in industrial-scale farms, as they are in the West, but their body parts are central to Chinese Traditional Medicine, which uses them for treating disease, often without any evidence of efficacy. Animal welfare as a concept was only introduced to China from the West in the last two decades [12]. It is not enshrined in Chinese law to anything like the extent to which it is in the West, nor is it well recognized by the public. Even today, most people in China are not familiar with the term ‘animal welfare’ [20,21,22], and there is no direct translation of this term from English into Mandarin. Young people, those who are better educated, and those in the eastern regions are most likely to be aware of the concept [20]. If people have heard of it, it is usually in relation to terrestrial animals; there is less understanding of the concept of fish welfare [23]. Women are more supportive of marine life protection than men, however [24]. Nevertheless, fish welfare research has commenced in China and has been focused on the transport of fish, which usually comprises live animals in tanks [25,26].

There is also a growing activist focus on animal welfare in China [27], in parallel with the burgeoning national and international scientific interest, at least, in the welfare of intensively farmed animals in China [18]. The Chinese public is divided on whether intensive production systems are a good thing or not [20]. Most of the internal literature is not accessible to Western academics and is directed toward improving the sustainability, biosecurity, and productivity of agricultural systems rather than improving the welfare of animals for their sake [18].

This has created a paradox for Chinese people, in which Chinese traditional teaching on the need to treat animals with respect has created a desire to tame nature and unite it with humankind [28]. However, recently, there has been an intensification of farm animal production on an enormous scale. This country has witnessed some of the most momentous transformations of nature on the planet, leaving diminishing resources for wildlife. Furthermore, the historical tendency of Chinese people in rural areas to eat wildlife led to the depletion of some rare species of birds and other animals [29]. During the famines of the 20th century, many Chinese people had to rely in part on wildlife for their food, and the acceptability of such practices into the 21st C has driven several species to extinction as the human population grows. It is a simple transition from wild animal biomass to human biomass, and often there is insufficient room for both. Another inevitable result has been a rise in the number of zoonotic diseases to which the human population is exposed and an increase in the impact of transmissible diseases, such as avian influenza, on wildlife populations.

Extending Chinese people’s utilitarian and instrumental attitude towards animals to meat consumption and the welfare of farmed animals, it is evident that good welfare is generally believed to equate with good product quality [22]. Stakeholders in the meat industry sometimes express reservations about this, believing meat quality to be primarily determined by breed and nutrition rather than welfare [30]. Nevertheless, the public may be prepared to pay more for food from animals with good welfare because they believe the quality to be better. Quality here includes food safety, which is considered more important by Chinese stakeholders than, for example, stakeholders from Malaysia, Thailand, or Vietnam [30]. This attitude is also likely to prevail outside of Asia. Chinese people are less likely than those in other BRICS countries to want to pay more for meat to increase animal welfare [31].

The Chinese people consider caring for wildlife more important than other animals [22]. A cross-cultural survey of students in 11 Eurasian countries found that although Chinese students had the least respect for animal welfare of students in any country, they had relatively high levels of respect for wildlife and accepted the use of animals as spiritual symbols more than other countries [32]. One of the reasons for limited support for animal welfare in China was the students’ limited ability to purchase high-welfare products in China, as they had the least expenditure of students in any country. Thus, their ability to pay was in question, even though they attributed several animal species with high levels of sentience. When considering the general public rather than students, a greater willingness to pay for animal products from farms with high welfare standards is evident. Pork consumers were willing to pay 5–23% more for high-welfare pork products in a recent survey [33]. Despite this, high meat prices relative to expenditure have resulted in poultry, beef, mutton, and fish being considered luxuries in many Chinese households [34]. It is not that Chinese people do not recognize animals’ needs, but for some, their limited financial resources make it hard for them to take actions that improve animal welfare. For example, the availability of equipment to render animals unconscious before slaughtering them for food is scarce in China, which is a major welfare issue [13]. Provision of such resources, together with training in the use of the equipment, are vital steps in making Chinese animal production more internationally acceptable. There is also a perception in China that stunning reduces meat quality, which may derive from experiences when it is done badly [35]. Done well, stunning improves meat quality [35].

Evidence that communism, as a government policy, has had an influence on attitudes to animal welfare is also provided by recent surveys because the attitudes of students from various communist and former communist countries are similar to each other and different from attitudes in capitalist countries [32,35]. Communism emphasizes egalitarian approaches to the workplace, so women are frequently in charge of looking after animals. Their naturally more empathetic approach to animals compared to men [36] should help to support good animal welfare, provided that the production system allows it.

The Chinese government has been paying increasing interest to animal welfare, as evidenced by the increase in funded research projects over the last three decades [37]. College students are emerging with an improved understanding of animal welfare, with the greatest concern for companion and wild animals, and less for farm and laboratory animals [38]. Animal scientists are aware of animal welfare issues and government regulations have been developed, but implementation is limited [39].

## 5. Attitudes of People Working in the Livestock Industries, Especially Towards Slaughter of Farm Animals

Most Chinese workers in the poultry industry have a reasonable understanding of animal welfare concepts; however, cage-free systems are only supported if there is an economic incentive to do so [40]. Many suppliers have negative attitudes towards cage-free egg production because of concerns about hen health, productivity, food safety, and profitability [40]. However, free range is a recognized concept in relation to hen welfare, which may encourage people to purchase the product on account of better sensory characteristics of the products [41].

In the dairy industry, workers make a strong connection between human and animal welfare, and improving the latter may be a route to improving the former [42]. The organizational culture of a farm is critical to engendering a compassionate approach to animal welfare [43]. On most farms, workers are willing to give pain relief to cattle experiencing some of the most painful conditions, such as dystocia, C-section, and calving [44].

Compared to nearby countries in SE Asia, such as Thailand, Malaysia, and Vietnam, Chinese stakeholders in the livestock industries are more likely to accept the killing of animals than those from Thailand, and they find it more acceptable for animals to experience pain and suffering during slaughter than those from Malaysia [45]. Nevertheless, killing animals without stunning is believed by stakeholders in the livestock industries to be a major welfare issue, and most Chinese people believe that animals should be rendered unconscious before they are killed [22]. Peer pressure to improve animal welfare appears less in China than in neighboring countries, with the result that Chinese stakeholders in the livestock industries are less likely than those in Malaysia, Thailand, and Vietnam to agree that people who are important to them would approve of them making improvements to animal welfare, that they intend to make improvements in the welfare of animals, and that they have confidence in their ability to make these improvements [46]. This is because they believe there are more pressing community issues. If they do support animal welfare improvements, it is most likely because there were monetary benefits to the community (reflecting their communal, rather than individual approach to societal living), not because of the morality of doing so. Similarly, benefits to their company, rather than themselves, are rated highly as motivators for individuals to improve animal welfare [30]. However, doubt that improving animal welfare will bring financial benefits persists among stakeholders in the livestock industry in China [30]. Nevertheless, the benefits of good animal welfare to the workers on dairy farms are one of the hidden benefits that make dairy farming a more satisfying occupation. Animals provided with good welfare will be easier to handle, potentially growing faster and producing more milk, and if they have a good demeanor, it is likely to be extended to the workers [42].

## 6. Factors Influencing Chinese Attitudes to Animals

### 6.1. The Law

China has had a national law aimed at maintaining standards of good animal welfare since 1988 [47], but this is not widely recognized (see for example, [48]). There are standards of husbandry to cover farm animal husbandry, transport, slaughter, and disease control (Table 1) [49]. These are detailed prescriptions for the management of farm animals, including housing, stocking densities, prohibition of the use of electric goads, etc.

Urbanization, greater pet ownership, and overseas developments have increased people’s desire to have the treatment of animals regulated [47]. Nevertheless, the understanding and application of animal welfare law in China is limited. Stakeholders in the livestock industry believe that making and upholding legal recommendations for animal welfare in China is difficult [13]. Although the government has close relations with businesses in China, it is not likely to enforce improvements in animal welfare through the law, except in extreme cases [13]. This is largely because the Chinese government does not see animal welfare as of sufficient importance to require legislative control, which is in stark contrast to, for example, the European Union. Animal welfare law in the European Union does, however, influence production systems in China if export to the EU is the major aim of production [47].

The welfare of wildlife has been significantly impacted by the COVID pandemic, which dramatically changed people’s attitudes to wildlife and, in particular, their consumption, which was previously commonplace even in the cities [62]. In 2019, the Chinese government passed legislation [63,64] banning wildlife consumption. In fact, change was already afoot: in 2016, the Chinese government passed a law banning cruel hunting methods, including poisons, explosives, snares, leg-hold traps, and night-time hunting with lights [65]. The Chinese government has also introduced policies to increase people’s awareness of nature, including a ‘Bird-Loving Week’ [66].

### 6.2. Type of Production System

A recent report suggested that most broiler farmers in China recognize the importance of animal welfare and are willing to implement changes to improve animal welfare on their farms [67]. Stakeholders in the Chinese pig and poultry industries who run free-range units were more likely to believe that they could make improvements to the welfare of animals in their care than those with intensive indoor systems [13]. This demonstrates the relative inflexibility of the latter, although free-range farmers may also have had greater intention to make changes. Farmers operating large production systems (>10,000 birds) are less likely to want to make improvements, compared with smaller units [13], probably because of the difficulties involved. Large, intensive production units are therefore inflexible when it comes to welfare improvement. Pigs are commonly seen as more intelligent than chickens, which may encourage farmers to want to provide for their welfare [13] and the government to support more studies of pig than poultry welfare [18]. Stakeholders in these industries generally believe that their animals should be physically healthy and happy, that they feel pain, suffer, and should be allowed to nest before parturition. Farm animals are seen as intelligent, friendly, understanding of their environment, and resentful of routine procedures such as tail docking and teeth clipping [13]. This attitude of the stakeholders is clearly at odds with existing provisions for most farm animals. Many practices that have been outlawed in parts of the West, such as keeping chickens in battery cages, toe clipping, forced molting, and keeping sows in stalls for long periods, persist in China. Hence, it is not surprising that many foreigners have a poor opinion of the welfare of farm animals in China [68].

Not having the tools and resources to make improvements is a significant impediment for those working directly with animals [69]. Farmers are particularly strongly motivated by the attitudes of their peers, so engendering change in groups of farmers, rather than individuals, is highly desirable [70].

### 6.3. Gender

In a survey of students in China and other Eurasian countries, women were credited with more benign attitudes towards animals’ welfare than men, although this is not necessarily true if the women are in situations where they are subservient to, and must adopt, the male household leader’s attitudes [71]. However, if women are free to express their opinions, they are naturally more empathetic, which may derive from their greater role in the nurturing of children or, in primitive society, their caring for animals in the homestead, when men went hunting animals. Despite this and the hierarchical society in China, women there have expressed greater concern for animals’ welfare than men, though not to the extent of highly emancipated Western countries [71]. However, concern for animal protection has also been rated similarly by Chinese men and women [46], suggesting that the gender difference exists for the welfare but not the rights of animals.

Two recent studies suggest that the difference in attitudes between men and women in China is age-dependent, with younger females having more benign attitudes than older males [24,36]. Women tend to see animals as within their social group, whereas for men, they are often outside of it. In one survey, people who chose not to reveal their gender expressed even more sensitivity to animal welfare [24].

### 6.4. Age, Experience, and Occupation

Young people in China demonstrate a more positive attitude toward animals than middle-aged or old respondents [72,73]. They are also more likely to avoid the use of bear bile in medicines than older people [74]. However, in another survey, older people were more likely to agree that animals used for food should be well cared for than younger ones [75]. Those with better education also had a more positive attitude [73]. Older people with more experience in the pig and poultry industries are more likely to recognize that the welfare of their animals is important; however, younger people are more likely to try to improve welfare [13]. This demonstrates a greater sensitivity to welfare that comes with age, but a greater willingness in young people to make changes. When it comes to the ancillary industries, government officials are more likely to respect good animal welfare than those working directly with animals. Veterinarians are more likely to recognize poor welfare but less likely to think people want it improved, compared with those working directly with the animals [13,70]. Amongst college students, Miao et al. identified an inverse relationship between support for vegetarianism, animal rights, and welfare on the one hand and a belief in a scientific approach to wildlife management on the other. College students mostly thought that trophy hunting was cruel, which is perhaps not surprising as it has been banned since 2006 [66].

### 6.5. Political Ideology

People with absolutist (idealist) attitudes show the most positive attitude toward animals, compared with those with more subjective attitudes [73]. Idealists were also positively disposed towards animals, whereas those with a relativistic attitude were more negatively disposed [72,73]. The COVID pandemic appears to have influenced Chinese people’s attitudes, with greater concern after the pandemic than before, which has been attributed to a less idealistic approach.

### 6.6. Religion

Although about two-thirds of Chinese people do not follow any particular religion nowadays [32,75], the remaining third are likely to have their attitudes influenced by the religious doctrines that they follow. Even if they do not follow a religion, Chinese people are likely to have been influenced by the religious beliefs of their ancestors. The earliest religious influences on Chinese people were those of Daoism, founded in the 6th century BCE, but having its major influence in the Tang, Song, and Yuan dynasties from 618–1368 CE. Confucianism was also followed in the Tang period and peaked in the Song period, but it was not extensively followed in the Yuan dynasty. Most scholars accept that attitudes in Chinese society have been shaped by 2000 years of Daoist and Confucian teaching. Whereas Daoism emphasizes submissiveness, meekness, and empathy for others, including animals, Confucianism emphasizes respect for elders and authority. These are the yin and yang of Chinese society, respectively. Daoism also respects nature and refutes any attempts to interfere with it. To illustrate this, the Daoist sage Zhuang Zi wrote, “A duck has short legs, but if we try to stretch them it will feel pain. A crane has long legs, but if we try to shorten them it will feel distress” [76]. In contrast to this, Confucius taught that humans mattered most, but animals had important roles in rituals and sacrifices, and, as a part of the country’s resources, were allowed to be used by humans as they felt appropriate [77]. His successor, Mencius, advocated a more empathetic attitude towards animals, recognizing that they have feelings but still affording them second place after humans. Confucius’ vision of equitable and harmonious living allowed society to develop relative prosperity and stability until the 20th century, when the momentous changes of that century brought about major changes in philosophy and practices towards animals.

About 1500 years ago, Buddhism flourished in China, bringing a much stronger sense of communion with nature and respect for native animals than was evident in Confucianism [19]. Although Daoism originally sanctioned meat consumption, Buddhism advocated a vegetarian lifestyle and a high level of respect for animals [77]. That this has influenced modern society was suggested in a survey of students in various Eurasian countries (China, Czech Republic, Great Britain, Iran, Ireland, South Korea, Macedonia, Norway, Serbia, Spain, and Sweden) [46]. Students from China rated the importance of concern for animals to protect the environment higher than those from most other countries. More recently, there was a rapid decline in the influence of religion on attitudes toward animals before and after the COVID pandemic [72]. Those following the Islamic faith are reported to have less concern about animal issues than those following other faiths [72].

Although stakeholders in the livestock industries in China deny that religion is a major influence on their attitudes toward animals [47], it is inevitable that the attitudes of society have been shaped by these religious influences. Nevertheless, Cao [19] argues that the Chinese have adopted a largely instrumental approach to animals, which are thought to be valuable only insofar as they are useful to humans. This position is normally associated with Western beliefs, underpinned by statements in the Christian Bible that man has dominion over animals on Earth. Thus, in China, hunting wild animals is generally tolerated despite its major impact on their populations, particularly birds [19]. In Confucianism and Daoism, overt cruelty to animals is of general concern, but using them for human benefit is supported. However, just as in the West, there is an abundance of literature, often focused on children, to instruct the readers that animals have morality and a sense of self. In China, this literature often depicts wild animals as fierce, ferocious, and sometimes friendly but always conveys the idea that animals live in a world of right and wrong. Such images, whilst helpful in encouraging young readers to adopt a responsible respect for animals, are meant to instruct readers on how to behave in a humane society.

## 7. Changing Attitudes

Changing attitudes is an important part of changing behavior. Consumers in the West have been demanding improvements in farm animal welfare, and it is likely that a similar call for improvements will occur following the intensification of the animal industries in China, which has been unchecked in the last decade or so. Animal welfare knowledge can be improved, at least temporarily, if stakeholders in farm animal production are instructed in animal welfare [75]. Those in the dairy sector absorb animal welfare concepts and information more readily than those in the sheep sector [75], suggesting that the latter may find it harder to embrace a novel, zoocentric approach to animal husbandry. Instruction to farmers on improved welfare techniques directly on demonstration farms has been difficult in the context of growing concerns about on-farm biosecurity [78]. Demonstration farm videos are valuable to educate farmers remotely, but they may not always invoke change in attitudes [78]. For the consumer, attaching information on animal welfare to food labels can help to reduce the consumption of meat products, an objective of the Chinese government [79].

## 8. Discussion

Attitudes to animal welfare in China are uniquely influenced by the development of Chinese culture, from the historical influences of Confucianism, Daoism, and Buddhism to the more recent influences of communism. Coupled with this has been the major influence of the rapid economic development of the Chinese economy in recent times and the focus on increasing prosperity, including access to inexpensive animal products. Hence, the attitudes of most people demonstrate a pronounced paradox, more so than in Western countries, in which the beliefs of the people in providing good welfare for animals, and especially wildlife, that are managed by humans is not matched by concern for the systems used to farm animals. In response, the Chinese government has enacted legislation aimed at ensuring good systems of welfare, but compliance mechanisms are not assured, particularly when adherence to moral ideals is a communal responsibility, not necessarily enforced by law. As China is a major exporter of animal products, welfare standards in countries importing their products are driving some Chinese standards up. However, the Western belief in the superiority of free-range systems for pigs and poultry gains little support in China, where there is extensive competition for land and other resources needed by free-range systems. Cultured animal products could radically change the attitudes of the Chinese people towards animals [80], along with land released for wildlife, which the Chinese people so badly want to preserve. Instrumentalism may give way to a greater reverence for animals, respecting the Daoist traditions in China that had begun to be eroded in modern society.

Similarly, the high human and animal population densities in China and the risks to food and feed security that this brings may encourage a return to primarily plant-based agriculture, which was so prevalent through most of the twentieth century. The Chinese government is advocating restraints on meat consumption that may support this; in 2016, it targeted a 50% reduction in meat consumption by 2030 [81]. Traditionally, tofu and other soy-based products have been readily available in China, and renewed utilization of these in the Chinese diet could help to change attitudes to make wildlife the focus of animal welfare concerns. This is likely to be different in the various regions of China, given the major variations in diet that currently exist, from a reliance on cattle and sheep in the northern regions to a more varied diet with many plant and animal products in the south and a greater reliance on fish in the eastern, coastal regions.

## 9. Conclusions

Historically, Chinese people were taught to be beneficent towards animals, in their culture, religion, and mythology. In the modern era, there is a more utilitarian attitude towards farm animals, but farm animal welfare is still valued for its contribution to product quality. Attitudes towards wildlife have changed rapidly, especially since the COVID pandemic, and they are becoming valued in their own right rather than for consumption. The Chinese government has introduced legislation that charts the path toward more humane production methods for farm animals. This is accompanied by a growing interest in animal welfare by the Chinese scientific and advocacy communities.

## Figures and Tables

**Table 1 animals-15-00878-t001:** National standards for the husbandry of animals in China, from [50].

Period	Species	Standard No. [reference]	Title
Rearing	Pig	GB/T 17824.3-2008 [50]	Environmental parameters and environmental management for intensive pig farms
		GB/T 20014.9-2013 [51]	Good agricultural practice—Part 9: Pig control points and compliance criteria
		GB/T 32149-2015 [52]	Specification for clean production of intensive pig farms
	Dairy	GB/T 37116-2018 [53]	Feeding specification for dairy replacements
	Cattle and Sheep	GB/T 20014.7-2013 [54]	Good agricultural practice—Part 7: Cattle and sheep control points and compliance criteria
	Poultry	GB/T 20014.10-2013 [55]	Good agricultural practice—Part 10: Poultry control points and compliance criteria
		GB/T 32148-2015 [56]	Specifications for healthy poultry production
Transport		GB/T 20014.11-2005 [57]	Good agricultural practice—Part 11: Livestock transport control points and compliance criteria
Slaughter		GB/T 22569-2008 [58]	Technical specification for humane slaughtering of pigs
		GB/T 17236-2019 [59]	Operating procedures of livestock and poultry slaughtering—Pig
		GB/T 19479-2019 [60]	Good manufacturing practice for livestock and poultry slaughter—Pig
Killing for disease control purposes		GB/T 42071-2022 [61]	Welfare during killing animals for disease control purposes

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
