# Peer review of "Developments in Chinese Attitudes to Animal Welfare"

_animals, 2025, doi:10.3390/ani15060878_

Round 1
Reviewer 1 Report
Comments and Suggestions for Authors
This manuscript presents itself as a literature review of the topic, but the “Methods” section is reflects just basic literature searching. Several of the major sources of potential citations are not mentioned , e.g. SciLit, Scopus. The methods section does not follow the usual depth of detail about articles identified and retrieved, excluded from review and organization by content analysis.
The paper includes several vague and unreferenced characterizations of Chinese society, e.g. lines 54-55 “Chinese people tend to have more of a sense of long-term orientation and engage less in impulsive reactions”, line 85 “not able to overcome their desire to tame nature” and line 80 “The tendency of Chinese people in rural areas to want to eat wildlife”. In the absence of documentation, these statements come off as stereotyping or even racist.
Several mentions are made of the inadequacy of Chinese law pertaining to animal welfare. (line 68, lines 208-219, line 334) yet there are no specifics given about the actual laws in place, their provisions as they relate to wildlife or agricultural animals or any discussion of the barriers to enforcement.
The discussion of concerns about zoonotic diseases associated with reliance on wildlife as food makes no mention of COVID and other SARS related diseases, which is a significant omission.
The statement on line 338 “the production of cellular meat and synthetic milk are likely to be strongly supported” is naïve and completely out of touch with Western reality which, in the U.S. has taken the form of widespread adoption of outright statewide bans or restrictive labeling of cellular meat and even non-animal "milks", particularly in areas with strong dependence on the kind of conventional industrial scale agriculture gaining ground in China. The author’s lit review should pay closer attention to the vast contemporary literature on the “failed promise of cellular meat.”
Author Response
Reviewer 1
This manuscript presents itself as a literature review of the topic, but the “Methods” section is reflects just basic literature searching. Several of the major sources of potential citations are not mentioned , e.g. SciLit, Scopus. The methods section does not follow the usual depth of detail about articles identified and retrieved, excluded from review and organization by content analysis.
Author’s response: I have added information on the number of hits from the Web of Science, the principal research engine used.
The paper includes several vague and unreferenced characterizations of Chinese society, e.g. lines 54-55 “Chinese people tend to have more of a sense of long-term orientation and engage less in impulsive reactions”,
Author’s response: Thank you for pointing this out, I have added that this statement derives from work by Hofsted, referred to in the second sentence of this paragraph.
line 85 “not able to overcome their desire to tame nature”
I am sorry, this was indelicately put. I have revised this early section of the paragraph to more accurately reflect the paradox for Chinese people, which Deborah Cao recognised in her seminal work.
and line 80 “The tendency of Chinese people in rural areas to want to eat wildlife”. In the absence of documentation, these statements come off as stereotyping or even racist.
Author’s response: It was definitely not meant to be racist, because the same historical tendency has been observed in many countries. I also explain that the historical consumption of wildlife was out of necessity. However, this statement has been revised with references to reflect the fact that attitudes have been changing rapidly in this respect. It is also referred to later (Factors influencing Chinese attitudes to animals: The Law).
Several mentions are made of the inadequacy of Chinese law pertaining to animal welfare. (line 68, lines 208-219, line 334) yet there are no specifics given about the actual laws in place, their provisions as they relate to wildlife or agricultural animals or any discussion of the barriers to enforcement.
Author’s response: I have now expanded this section to include a list of extant laws addressing the husbandry of farm animals, which include many welfare considerations (Table 1).
The discussion of concerns about zoonotic diseases associated with reliance on wildlife as food makes no mention of COVID and other SARS related diseases, which is a significant omission.
Author’s response: a paragraph on the impact of Covid on wildlife consumption, including the law, has been added at the end of the section on Factors influencing Chinese attitudes to animals: The Law. I have removed wildlife consumption from the conclusions, taking this to be more historically important than today.
The statement on line 338 “the production of cellular meat and synthetic milk are likely to be strongly supported” is naïve and completely out of touch with Western reality which, in the U.S. has taken the form of widespread adoption of outright statewide bans or restrictive labeling of cellular meat and even non-animal "milks", particularly in areas with strong dependence on the kind of conventional industrial scale agriculture gaining ground in China. The author’s lit review should pay closer attention to the vast contemporary literature on the “failed promise of cellular meat.”
Author’s response: I have added two sentences to this discussion, which includes associated references to studies undertaken in China to investigate the public attitude to cultured animal products, which shows diverse responses. I appreciate this field is changing very fast and am happy to remove this part if necessary.
Reviewer 2 Report
Comments and Suggestions for Authors
In the abstract, I miss a sentence that provides some guidance on the conclusions reached.
The introduction is very short. I think should outline the order that the document will follow. Some points that are later developed are mixed in, but some content that will be explained later is missing. Additionally, there is a lack of organization.
On line 37, it directly discusses meat consumption, referring to land-based livestock production. It does not give the impression of including aquaculture, which is mentioned later.
The citation on line 38 and 39 does not give the impression of being from academic database sources, so I would refer to the original citation.
The introduction does not mention the species involved in animal production, nor does it clarify that it will later refer to wild animals. As a result, the direction of the animal welfare review is not clear. A recommendation for the introduction is to include a table with the variables that will be discussed for the analysis of animal welfare.
Historical context: In line 49, citation number 3 could be expanded by including the original article, presenting data to support this part.
In line 56, the citation is missing. I would recommend discussing the five freedoms of animal welfare and how they have influenced in Chinese science and politics.
In line 72, I miss a more recent citation and some information about specific legislation.
In this section, I would suggest marking the order with subtitles to clearly organize the section.
In the section about factors influencing Chinese attitude towards animals, I would recommend providing more detailed analysis and show this part with a table that gives solidity to the data.
In line 260, discussing only pigs and poultries, creates confusion. If it were put into more context, it would be clearer.
The discussion is brief. As I mentioned earlier, if there were a table that showed or guided toward the results, it would be more accessible and easier to understand. I miss that the method section does not mention alternative food sources, such as cellular meat and synthetic milk.
I would create a specific section for conclusions.
Author Response
Reviewer 2
In the abstract, I miss a sentence that provides some guidance on the conclusions reached.
Author’s response: there was a concluding sentence: ‘It is concluded that attitudes are changing rapidly, but so are animal production systems, which makes the development and improvement of animal welfare on farms especially important.’
The introduction is very short. I think should outline the order that the document will follow. Some points that are later developed are mixed in, but some content that will be explained later is missing. Additionally, there is a lack of organization.
Author’s response: I have expanded the Introduction and added at the end a section outlining the organisation of the manuscript. Thank you for that suggestion. The Introduction was short, but please remember that much of the historical context that follows is essentially introductory.
On line 37, it directly discusses meat consumption, referring to land-based livestock production. It does not give the impression of including aquaculture, which is mentioned later.
Author’s response: A statement on the dominance of China in aquatic products is now included, with reference to statistics. ‘It is also the largest producer of aquatic products in the world, producing 35% of the global output [3]’
The citation on line 38 and 39 does not give the impression of being from academic database sources, so I would refer to the original citation.
Author’s response: the statistics come from Statista, a German-based information source. According to a media bias factcheck website (mediabiasfactcheck.com/statista/), it is highly reliable, with high credibility.
The introduction does not mention the species involved in animal production, nor does it clarify that it will later refer to wild animals. As a result, the direction of the animal welfare review is not clear. A recommendation for the introduction is to include a table with the variables that will be discussed for the analysis of animal welfare.
Author’s response: The final section of the Introduction provides a precis of the contents of the review, including a final sentence of the types of animals included in the review.
Historical context: In line 49, citation number 3 could be expanded by including the original article, presenting data to support this part.
Author’s response: I have added a single sentence at this point and reference to the later discussion of the teachings of Confucius et al, under Religion.
In line 56, the citation is missing. I would recommend discussing the five freedoms of animal welfare and how they have influenced in Chinese science and politics.
Author’s response: the citation has been added. Discussion on the relevance of the Five Freedoms in China today has been added.
In line 72, I miss a more recent citation and some information about specific legislation.
Author’s response: a recent reference is included at this point {Zhou, 2024]. Information about specific legislation has been added in the first section of factors influencing attitudes to animal welfare, on the Law. This includes a table of current legislation.
In this section, I would suggest marking the order with subtitles to clearly organize the section.
Author’s response: Headings have been expanded and sections reorientated to make a more logical flow and section organisation.
In the section about factors influencing Chinese attitude towards animals, I would recommend providing more detailed analysis and show this part with a table that gives solidity to the data.
Author’s response: I have expanded this section and included a new subheading of political ideology. A table is included under Law.
In line 260, discussing only pigs and poultries, creates confusion. If it were put into more context, it would be clearer.
Author’s response: Unfortunately, the data only refers to people with experience of the pig and poultry industries.
The discussion is brief. As I mentioned earlier, if there were a table that showed or guided toward the results, it would be more accessible and easier to understand. I miss that the method section does not mention alternative food sources, such as cellular meat and synthetic milk. I would create a specific section for conclusions.
Author’s response: Discussion and Conclusions have been separated, as recommended, and a comprehensive Conclusion written. The comments on cellular meat have been moved to the Introduction and I now mention alternative sources of animal products in the outline of the layout of the article, at the end of the Introduction.
Round 2
Reviewer 1 Report
Comments and Suggestions for Authors
Thank you for incorporating most of the changes suggested.
Reviewer 2 Report
Comments and Suggestions for Authors
After the review, it is fine.